# Outpatient Antibiotic Resistance Patterns of *Escherichia coli* Urinary Isolates Differ by Specialty Type

Lauren Frisbie,[a] Scott J. Weissman,[b] Hema Kapoor,[c] Marisa D'Angeli,[d] Ann Salm,[c] Jeff Radcliff,[c] Peter Rabinowitz[a]

[a]Department of Environmental and Occupational Health Sciences, Center for One Health Research (COHR), University of Washington School of Public Health, Seattle, Washington, USA
[b]Infectious Diseases Department, Seattle Children's Hospital, Seattle, Washington, USA
[c]Infectious Diseases/Immunology, Quest Diagnostics, Secaucus, New Jersey, USA
[d]Washington State Department of Health, Shoreline, Washington, USA

**ABSTRACT** Antibiotic-resistant *E. coli* infections represent a major cause of morbidity and mortality and pose a challenge to antibiotic stewardship. We analyzed a large outpatient data set of *E. coli* urinary isolates to determine whether resistance patterns vary between types of outpatient practices. Using deidentified data from a clinical reference laboratory over 5 years and logistic regression, we examined the association of antibiotic resistance with outpatient practice type, controlling for testing year, patient sex, and patient age. The odds of antibiotic resistance were significantly higher in urology/nephrology practices for ampicillin (odds ratio [OR] 1.36; 95% CI, 1.10 to 1.69), ciprofloxacin (OR 2.29; 95% CI, 1.77 to 2.94), trimethoprim-sulfamethoxazole (OR 1.52; 95% CI, 1.18 to 1.94), and gentamicin (OR 1.72; 95% CI, 1.16 to 2.46). Odds of resistance were also higher for ciprofloxacin in oncology practices (OR 1.54; 95% CI, 1.08 to 2.15) and "all other specialties" (OR 1.33; 95% CI, 1.13 to 1.56). In contrast, specimens from obstetrics and gynecology practices had lower odds of having resistance to ampicillin (OR 0.90; 95% CI, 0.82 to 0.99) and trimethoprim-sulfa (OR 0.83; 95% CI, 0.73 to 0.93) but higher odds of having resistance to nitrofurantoin (OR 1.33; 95% CI, 1.03 to 1.70). Other findings included lower odds of having resistance to trimethoprim-sulfa in pediatric practices (OR 0.78; 95% CI, 0.64 to 0.94) and lower odds of having resistance to gentamicin in isolates from internal medicine practices (OR 0.66; 95% CI, 0.51 to 0.84) (all $P < 0.05$).

**IMPORTANCE** Patterns of antibiotic resistance in *E. coli* urinary isolates can vary between outpatient specialties. The use of clinical data to create practice and specialty-specific antibiograms in outpatient settings may improve antibiotic stewardship.

**KEYWORDS** *E. coli*, UTI, antimicrobial resistance, outpatient, specialty

Antimicrobial resistance is a global and growing threat to public health, requiring ongoing appraisals of the use of antibiotics in different settings of medical care (1, 2). Although antibiotic-resistant pathogens are an increasing challenge to the care of hospital inpatients, more than half of antibiotic use in human health care occurs in outpatient settings. While antibiotic stewardship efforts in inpatient care have increased in recent years, stewardship efforts in outpatient settings remain less developed. It is increasingly recognized that outpatient use of antibiotics drives community resistance patterns (3). In a recent study of urinary tract infections (UTI) due to *Escherichia coli* (EC) in Washington State over 5 years, associations were found between resistance patterns, age, sex, and period among specimens tested (4). In addition, the frequency of antibiotic use differs between types of health care providers. A 2018 CDC report on oral antibiotic prescribing found primary care physicians to have 376 antibiotic prescriptions per 1,000 persons, compared to 137 per obstetrics and gynecology (Ob/Gyn) provider and 274 per year for all provider types (5).

Address correspondence to Lauren Frisbie, lfrisbie@uw.edu, or Ann Salm, Ann.E.Salm@QuestDiagnostics.com.

The authors declare a conflict of interest. Hema Kapoor, Jeff Radcliff, and Ann Salm are employed by and own stock in Quest Diagnostics.

**TABLE 1** Distribution and demographics of specialty categories

| Clinical specialty | No. (%) facilities | No. (%) isolates | Mean (SD) age, y | Female, no. (%) |
|---|---|---|---|---|
| General family practice | 340 (46.3) | 17,252 (71.2) | 48 (21.7) | 16,079 (93.2) |
| Internal medicine | 61 (8.3) | 1960 (8.1) | 65 (17.6) | 1728 (88.2) |
| Pediatrics | 26 (3.5) | 878 (3.6) | 10 (6.7) | 838 (95.4) |
| Obstetrics and gynecology[a] | 84 (11.4) | 2114 (8.7) | 38 (16.1) | 2101 (99.4) |
| Urology/Nephrology | 20 (2.7) | 357 (1.5) | 66 (15.3) | 317 (88.8) |
| Oncology | 14 (1.9) | 224 (0.9) | 67 (12.6) | 201 (89.7) |
| All other specialties | 190 (25.9) | 1430 (5.9) | 51 (21.3) | 1298 (90.8) |
| Total | 735 (100) | 24,215 (100) | 48 (22.6) | 22,562 (93.2) |

[a]13 males were identified in Ob/Gyn clinics; the data only included a binary sex gender classification of male and female, so no further conclusions on gender can be made.

One of the most common outpatient uses of antibiotics is to treat urinary tract infections, the majority of which are caused by *E. coli*. Even within this category, patterns of UTIs and approaches to antibiotic stewardship vary across medical care settings (6, 7) considering the "patient's situation, antibiotic resistance within each local community, treatment costs, and treatment failure rates" (8).

Based on the type of facility specialty, a health care setting will see patients differing by demographic characteristics and needs. The National Ambulatory Medical Care Survey (NAMCS) outlined some of the differences between specialty practices. For example, patients of oncology clinics, in addition to being immunocompromised, tended to be older than patients in some other specialties according to 2010 NAMCS data, with 88% above age 45 (9). In contrast, NAMCS data indicate that 69% of pediatric practice visits are for children younger than 10 years old. In addition to age differences, sex distributions vary across practice types. For example, Ob/Gyn practices focus on women's health, and urology/nephrology (uro-neph) practices have mostly male patients. These demographic differences could lead to differences in antibiotic resistance between specialties (10).

Although culture and sensitivity results provide the most accurate information about the resistance of a particular pathogen, such results take days, and initial treatment, therefore, is typically empirical. The Infectious Diseases Society of America (IDSA) recommends that empirical regimens for uncomplicated UTIs be guided by local susceptibility patterns (11). Yet, compared to hospital-based antibiograms, there traditionally has been less routine analysis of community antibiotic resistance patterns in the outpatient setting. Tracking and analyzing aggregate resistance patterns by practice setting, such as by specialty practice type, would allow customized antibiograms for more precise empirical antibiotic treatment decisions.

To assess whether differences exist in antibiotic resistance across outpatient care settings, we retrospectively analyzed deidentified antibiotic susceptibility test results for outpatient urinary *E. coli* isolates from a large clinical reference laboratory.

## RESULTS

**Specialty practices and isolates.** The data set contained 24,215 *E. coli* isolates from urinary samples collected over 5 years (Table 1) from 735 facilities. The largest number of *E. coli* isolates occurring among seven facility categories was general family practice (340 facilities, 46% of practice categories), followed by Ob/Gyn (84; 11%), internal medicine (61; 8%), pediatrics (26; 4%), uro-neph (20; 3%), oncology (14; 2%), and all other specialties (190; 26%). The other specialties category included psychiatry, endocrinology, gastroenterology, rheumatology, orthopedic, dentistry, plastic surgery, ophthalmology, and assisted living. Each of these had a small number of isolates.

The general family practice category accounted for the majority (71%) of the total isolates and was used as the reference group for logistic regression models. Most of the remaining isolates were from Ob/Gyn (9%), internal medicine (8%), and "all other specialties" (6%) practice categories (Table 1).

Oncology had the highest mean patient age, followed by uro-neph, internal medicine, all other specialties, general family practice, Ob/Gyn, and pediatrics. Most isolates were from females, though the proportions varied across practice types (88% to 99%) (Table 1).

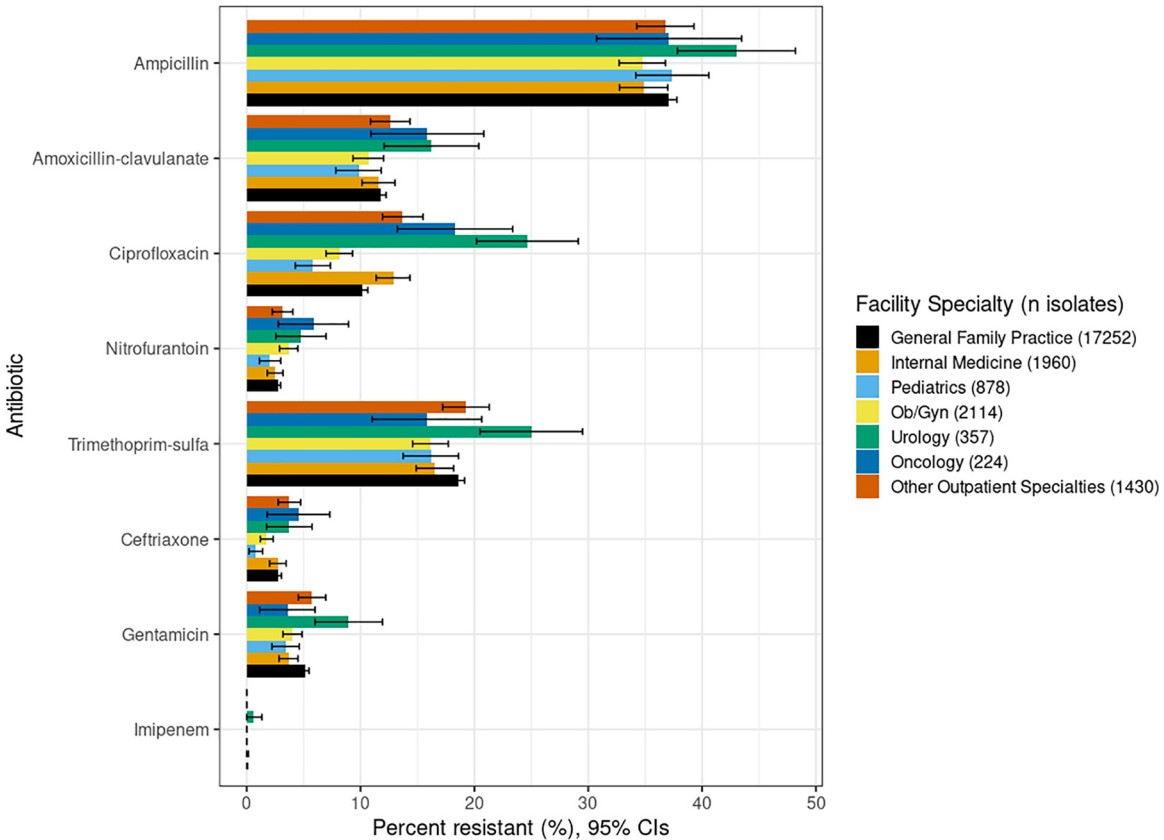

**FIG 1** Unadjusted rates of antibiotic resistance among urinary *E. coli* isolates by outpatient practice specialty.

**Antibiotic-specific resistance rates between facilities.** The highest resistance rates for the given agents were seen in different facilities. Isolates from uro-neph clinics had one of the highest percent of resistance for ampicillin (43%) ciprofloxacin (25%), gentamicin (9%), and trimethoprim-sulfa (25%). Isolates from oncology clinics had the highest percent resistance for ceftriaxone (5%) and nitrofurantoin (6%). Isolates from pediatric clinics had the lowest percent resistance for all antibiotics except for ampicillin (37%), where it was second highest. For all facility isolates, ampicillin showed the highest % resistance compared to the other antibiotics of interest (range, 34% to 43%) (Fig. 1 and Table S1).

**Multivariate associations of facility type and resistance.** After controlling for patient sex, age, and year of testing, we found several significant associations showing the odds of resistance for different antibiotics varied across facility types (Table 2). Compared to the reference group (general family practice), isolates from uro-neph clinics had higher odds of having resistance to ampicillin, ciprofloxacin, trimethoprim-sulfa, and gentamicin. Ob/Gyn clinics had higher odds of having resistance to nitrofurantoin. We found significantly higher odds of ciprofloxacin resistance in isolates from oncology clinics, as well as in isolates from the "all other specialties" category. In contrast, we found significantly lower odds of having resistance to both ampicillin and trimethoprim-sulfa among isolates from Ob/Gyn clinics, as well as higher odds of having resistance to trimethoprim-sulfa in isolates from pediatric clinics, and to gentamicin among isolates from internal medicine clinics.

## DISCUSSION

This analysis of a large data set of *E. coli* urinary isolates from outpatient medical practices identified significant differences in antibiotic resistance patterns between specialty types in Washington state that persisted after controlling for age, sex, and year of testing. These differences included higher odds of having resistance to multiple antibiotics among isolates from uro-neph practices, lower odds of having resistance to several antibiotics in pediatric

**TABLE 2** Adjusted[a]/multivariate association (odds ratio) between specialty and resistance for representative antibiotics

| Antibiotic[b] | Count, n | OR (95% CI) | P value | Antibiotic | Count, n | OR (95% CI) | P value |
|---|---|---|---|---|---|---|---|
| Ampicillin | | | | Trimethoprim-sulfa | | | |
| General family practice | 17139 | REF[c] | REF | General family practice | 17226 | REF | REF |
| Internal medicine | 1939 | 0.94 (0.85-1.04) | 0.253 | Internal medicine | 1954 | 0.90 (0.79-1.03) | 0.120 |
| Pediatrics | 872 | 0.97 (0.84-1.13) | 0.717 | Pediatrics | 878 | 0.78 (0.64-0.94) | **0.012** |
| Obstetrics and gynecology | 2098 | 0.90 (0.82-0.99) | **0.038** | Obstetrics and gynecology | 2113 | 0.83 (0.73-0.93) | **0.002** |
| Urology/Nephrology | 351 | 1.36 (1.10-1.69) | **0.005** | Urology/Nephrology | 356 | 1.52 (1.18-1.94) | **0.001** |
| Oncology | 221 | 1.05 (0.80-1.38) | 0.712 | Oncology | 221 | 0.86 (0.59-1.22) | 0.419 |
| All other specialties | 1414 | 1.00 (0.89-1.11) | 0.935 | All other specialties | 1428 | 1.05 (0.92-1.21) | 0.449 |
| Amoxicillin-clavulanate | | | | Nitrofurantoin | | | |
| General family practice | 16810 | REF | REF | General family practice | 17239 | REF | REF |
| Internal medicine | 1866 | 0.92 (0.78-1.07) | 0.264 | Internal medicine | 1960 | 0.74 (0.54-1.00) | 0.058 |
| Pediatrics | 855 | 0.89 (0.70-1.13) | 0.368 | Pediatrics | 877 | 0.83 (0.48-1.33) | 0.453 |
| Obstetrics and gynecology | 2032 | 0.93 (0.80-1.08) | 0.358 | Obstetrics and gynecology | 2114 | 1.33 (1.03-1.70) | **0.025** |
| Urology/Nephrology | 302 | 1.33 (0.96-1.80) | 0.076 | Urology/Nephrology | 357 | 1.19 (0.69-1.91) | 0.501 |
| Oncology | 208 | 1.31 (0.88-1.88) | 0.161 | Oncology | 222 | 1.63 (0.87-2.78) | 0.095 |
| All other specialties | 1404 | 1.06 (0.90-1.25) | 0.474 | All other specialties | 1429 | 1.11 (0.80-1.49) | 0.530 |
| Ciprofloxacin | | | | Gentamicin | | | |
| General family practice | 17252 | REF | REF | General family practice | 17252 | REF | REF |
| Internal medicine | 1960 | 1.00 (0.87-1.16) | 0.971 | Internal medicine | 1960 | 0.66 (0.51-0.84) | **0.001** |
| Pediatrics | 878 | 1.10 (0.81-1.48) | 0.522 | Pediatrics | 878 | 0.75 (0.50-1.08) | 0.139 |
| Obstetrics and gynecology | 2114 | 0.97 (0.82-1.14) | 0.736 | Obstetrics and gynecology | 2114 | 0.81 (0.64-1.01) | 0.070 |
| Urology/Nephrology | 357 | 2.29 (1.77-2.94) | **<0.00001** | Urology/Nephrology | 357 | 1.72 (1.16-2.46) | **0.005** |
| Oncology | 224 | 1.54 (1.08-2.15) | **0.015** | Oncology | 224 | 0.64 (0.29-1.23) | 0.225 |
| All other specialties | 1430 | 1.33 (1.13-1.56) | **0.0004** | All other specialties | 1430 | 1.11 (0.87-1.39) | 0.394 |
| Ceftriaxone | | | | | | | |
| General family practice | 17129 | REF | REF | | | | |
| Internal medicine | 1935 | 0.97 (0.71-1.29) | 0.828 | | | | |
| Pediatrics | 871 | 0.52 (0.22-1.04) | 0.093 | | | | |
| Obstetrics and gynecology | 2096 | 0.82 (0.57-1.14) | 0.251 | | | | |
| Urology/Nephrology | **347** | **1.82 (0.97-3.11)** | **0.042** | | | | |
| Oncology | 220 | 1.91 (0.93-3.48) | 0.053 | | | | |
| All other specialties | 1412 | 1.34 (0.99-1.78) | 0.047 | | | | |

[a]All models were adjusted for sex, age in years, and year of the test. Bold indicates significant association ($P < 0.05$).
[b]Antibiotic susceptibility panels differed between patients, resulting in different distributions of antibiotics per patient and facility type.
[c]General family practice was used at the reference group for logistic regression models.

practices, and lower odds of having resistance to nitrofurantoin in isolates from Ob/Gyn specialty practices.

Uro-neph clinic isolates had some of the highest rates of resistance compared to other specialties. uro-neph patients often include those with structural abnormalities in their urinary tract, as well as patients with recurrent UTIs. The average age of uro-neph patients in our study was also higher than the average among all specialties. Such patients are likely to have complicated and recurrent urinary tract infections, involving more frequent treatment that could be a driver for increased resistance. The 2015 NAMCS listed ciprofloxacin as a top active ingredient in prescriptions from uro-neph clinics, consistent with our finding of increased odds of resistance to this antibiotic among uro-neph isolates (12). There may also be uro-neph-specific antibiotic treatment guidance for urinary tract infections in the specialty clinic setting that promotes the use of broader spectrum antibiotics, such as fluoroquinolones. The 2019 American Urological Association Guidelines recommend clinicians "use first-line therapy dependent on the local antibiogram for the treatment of symptomatic UTIs" (13).

However, the availability and distribution of these local antibiograms will differ by the clinician. Typical guidelines are developed to be used by health care professionals across various departments and are not specialty-specific, highlighting the need for specialty-specific local antibiograms and antimicrobial stewardship interventions around education and the best empirical antibiotic choice (8).

Similarly, specimens from Ob/Gyn clinics had higher odds of having resistance to nitrofurantoin. The American College of Obstetricians and Gynecologists recommend sulfonamides

and nitrofurantoin to be prescribed in the first trimester of pregnancy only when other antimicrobial therapies are deemed clinically inappropriate (14). A CDC analysis of filled prescriptions found that among pregnant women with UTIs, nitrofurantoin, ciprofloxacin, cephalexin, and trimethoprim-sulfa are the most frequently prescribed antibiotics (15, 16). Urinary tract infections are more common among women than men (17, 18), so it is logical that resistance rates of nitrofurantoin might be higher in Ob/Gyn practices than in general practice settings (15).

This study builds off the previous work from Frisbie et al. (4) that looked at the associations between age and sex with antibiotic resistance patterns in the outpatient setting.

This study carries the same limitations as Frisbie et al. (4), including the bias introduced by excluding isolates from current infections, most patients being females, possible selection bias of the patient populations seeking care in outpatient settings, and lack of data on additional covariates, such as geography, socioeconomic status, insurance type and race/ethnicity (4). It is important to note that these data are a subset of the general population and includes a selection bias as not every patient's urine culture will be collected and subsequently have a susceptibility test done. This may lead to an overestimation of resistance rates (19).

In summary, the findings of this study suggest that antibiotic resistance in *E. coli* urinary isolates can vary across outpatient practice types which can inform treatment decisions. As part of the CDC's Antibiotic Resistance (AR) Solutions Initiative, one of the Centers for Disease Control's (CDC) activities is to "improve antibiotic use across health care settings, including telehealth, dental settings, outpatient settings, and STD clinics" and provide "evidence and tools for facilities to implement antibiotic stewardship practices and programs". The specialty-specific outpatient trends found in this analysis align with CDC's activities in fighting antibiotic resistance and support CDC's push to assist facilities (16). As data and studies on inappropriate prescribing practices for UTIs emerge (20), there is an urgent need for use of clinical data to create facility- and specialty-specific antibiograms in outpatient settings that may enable improved and "precise" antibiotic stewardship.

## MATERIALS AND METHODS

**Study design and population.** For this retrospective cross-sectional analysis, we analyzed data from a large clinical reference laboratory to assess antibiotic susceptibility tests of *E. coli* isolates collected from urinary sources in outpatient settings in Washington State from January 2013 to December 2017. As previously described (4), these data were available to the University of Washington without personal identifiers, under an academic-corporate research agreement (4). Results from patient antimicrobial susceptibility test (AST) results were included for analysis if they were from a urinary source and were collected in Washington State during the study period. We included the first isolate recorded for each patient during the 5 years, as recommended by the Clinical and Laboratory Standards Institute (CLSI) for analysis and presentation of cumulative antimicrobial susceptibility test data.

**Data analysis. (i) Classification of specialties.** The data included a variable detailing the type of facility where isolates originated, encompassing 34 facility types. We used a classification scheme based on the National Ambulatory Medical Care Survey (NAMCS) to classify each facility into one of seven categories: general family practice, internal medicine, Ob/Gyn, oncology, pediatric, uro-neph, and "all other specialties." This recategorization into seven groups allowed for larger group sizes and provided a reasonable way to look at resistance patterns across specialties to allow for more robust comparisons statistically because 14% ($n = 103$) of the original practice categories included fewer than five clinics per category.

**(ii) Antibiotic resistance.** We classified an isolate as "resistant" to a particular antibiotic if the results were interpreted as either resistant or intermediate according to 2017 CLSI standards (21). We focused on resistance results of eight different antibiotics representing different antibiotic classes (oral and injectable): penicillin (amoxicillin-clavulanic acid), penicillin (ampicillin), trimethoprim-sulfa, nitrofurantoin, cephalosporin (ceftriaxone), aminoglycoside (gentamicin), and quinolone (ciprofloxacin), and carbapenem (imipenem). We then compared resistance rates for these eight antibiotic classes across the seven facility types.

**(iii) Logistic regression.** Logistic regression was used to examine the association of outpatient practice type with antibiotic resistance, including covariates of the year of testing, patient sex, and patient age. This analysis was a continuation of the models used in Frisbie et al. (4), associations of antibiotic resistance with patient age, stratified by patient sex. All statistical models and analyses were created and performed in R version 3.6.3 (22).

**Ethical approval.** All study protocols were reviewed and approved by the Human Subjects Review Committee of the University of Washington.

## SUPPLEMENTAL MATERIAL

Supplemental material is available online only.

**SUPPLEMENTAL FILE 1**, PDF file, 0.03 MB.

## ACKNOWLEDGMENTS

This work was supported by a Memorandum of Understanding/Data Use agreement between Quest Diagnostics Clinical Laboratories Inc. and the University of Washington. The corresponding author received travel support from Quest Diagnostics to present research at IDWeek 2019.

Hema Kapoor, Jeff Radcliff, and Ann Salm are employees of Quest Diagnostics and own stock in the company. All other authors declare no conflict of interest.

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
