## [Reviewer comments · Microbiology Spectrum]

Microbiology Spectrum

Outpatient Antibiotic Resistance Patterns of Escherichia coli Urinary Isolates Differ by Specialty Type

Lauren Frisbie, Scott Weissman, Hema Kapoor, Marisa D'Angeli, Ann Salm, Jeff Radcliff, and Peter Rabinowitz

Corresponding Author(s): Ann Salm, Quest Diagnostics

Review Timeline:

Submission Date:	December 1, 2021
Editorial Decision:	January 27, 2022
Revision Received:	March 28, 2022
Editorial Decision:	April 4, 2022
Revision Received:	April 8, 2022
Accepted:	April 11, 2022

Editor: Ahmed Babiker

Reviewer(s): Disclosure of reviewer identity is with reference to reviewer comments included in decision letter(s). The following individuals involved in review of your submission have agreed to reveal their identity: Janet A Hindler (Reviewer #2)

Transaction Report:

DOI: <https://doi.org/10.1128/spectrum.02373-21>

January 27, 2022

Dr. Ann E Salm
Quest Diagnostics
Infectious Diseases/Immunology
500 Plaza Drive
Secaucus, NJ 07094

Re: Spectrum02373-21 (Outpatient Antibiotic Resistance Patterns of Escherichia coli Urinary Isolates Differ by Specialty Type)

Dear Dr. Ann E Salm:

Thank you for submitting your manuscript to Microbiology Spectrum. As you will see your paper is very close to acceptance. Please modify the manuscript along the lines recommended by the reviewers below. When submitting the revised version of your paper, please provide (1) point-by-point responses to the issues raised by the reviewers as file type "Response to Reviewers," not in your cover letter, and (2) a PDF file that indicates the changes from the original submission (by highlighting or underlining the changes) as file type "Marked Up Manuscript - For Review Only". Please use this link to submit your revised manuscript - we strongly recommend that you submit your paper within the next 60 days or reach out to me. Detailed instructions on submitting your revised paper are below.

Link Not Available

Sincerely,

Ahmed Babiker

Journals Department
Reviewer comments:

Reviewer #1 (Comments for the Author):

In their manuscript, "Outpatient Antibiotic Resistance Patterns of Escherichia coli Urinary Isolates Differ by Specialty Type," Frisbie and colleagues provide a comparison of susceptibility data between outpatient specialties that may commonly prescribe antibiotics for urinary tract infections. Their results (including more resistance in urology practices, more fluoroquinolone resistance in urology/oncology practices) may not surprise many Infectious Diseases clinicians, but is very helpful data to inform antimicrobial stewardship efforts and to help guide empiric antibiotic selection in such practices. The manuscript is clearly written, concise, and the figure and tables are helpful and complement the text well. Comments here are all minor and only aimed at potentially strengthening the manuscript.

Line 53 - I believe trimethoprim-sulfa would be considered an abbreviation for trimethoprim-sulfamethoxazole. Would write out the latter once.

Lines 75-77 - would use this sentence to point out that associations were found, rather than just assessed

Line 80 - Write out obstetrics and gynecology (Ob/Gyn) here rather than in line 96

Line 109 - Missing a word - perhaps exist? "To assess whether differences exist..."

Line 118 - Don't need to redefine Ob/Gyn abbreviation here

Line 126 - I think "other" here refers to the "All other specialties" category. Would keep terminology consistent.

Line 133 - "Change difference" to "different"

Line 139 - Typically the lower number in the range would be written first

Line 158 - Ob/gyn had higher odds of resistance for nitrofurantoin

Line 168 - If such urology-specific guidance exists, would cite it and point out how this data (and other data) suggests a need for revision

Lines 170-173 - I think (and some of my personal bias from stewardship experience may be showing here) you are being too generous to the Urologists here. If they were aware of prior resistance patterns, this and other data would suggest that ciprofloxacin should not be the top antibiotic prescribed from Urology offices (as you cited from the NAMCS data). A plausible speculation here would be whether the frequent choice of ciprofloxacin by urologists could be driving the development of resistance in their patients. It also seems to suggest a clear opportunity for antimicrobial stewardship teams to intervene with education around best empiric antibiotic choice.

Line 173 - US guidelines would be more relevant to cite here than Korean guidelines

Lines 182-183 - I believe the IDSA guidelines only discuss nonpregnant women

Line 187 - With regard to limitations, the decision to only include the first isolate and rationale for this is discussed in the authors' prior cited publication, however may carry greater weight in this study. For example, a patient may have a first urine culture sent from family medicine prior to referral to Urology, and then no urine cultures would be attributed to Urology over the subsequent 5 years, if I understand the methodology correctly. This may also potentially explain the relatively low number of isolates for Urology practices (including only 40 isolates from male patients from 20 practices over 5 years). As such, revisions to the methodology such as including one isolate per year per patient, including the first isolate per patient per specialty over the time period, or even including all isolates could be a consideration, to determine if it may be more informative about the within-specialty resistance patterns. At minimum, would articulate this limitation within the text of this manuscript rather than only referring to the prior one.

Line 198 - "CDCs" is missing an apostrophe

Table 1

--88.8% of isolates being from female patients for Urology practices seems quite high, given typically the large majority of patients in Urology practices are male. This seems to warrant a 2nd look to ensure the data is accurate / Urology practices are accurately categorized.

Table 2

--Why does the number of isolates vary between antibiotics? Were different susceptibility panels used? Would offer an explanation of this somewhere, assuming the numbers are accurate.

--There seems to be a discrepancy in the counting of isolates between this table and Table 1. For example, in Table 1, the total number of isolates for General family practice is listed as 17,252, but the number of isolates is higher than this for each antibiotic in Table 2. The count for "All other specialties" seems substantially lower in Table 2 compared to Table 1.

Reviewer #2 (Comments for the Author):

The authors analyzed data obtained from antimicrobial susceptibility testing at a large clinical reference laboratory to determine resistance rates for E. coli isolates from outpatients seen at 7 different types of outpatient clinical practices. They were interested in seeing if segregating antibiograms by practice might be useful for antimicrobial stewardship. Agents commonly used to treat urinary tract infections are included in the evaluation. The study, which "builds off" a previous study, is well done and easy to comprehend.

1. The authors indicate in line 187 that the "this study builds off previous work from the main author and carries the same limitations." These limitations are quite interesting, however, unless one has access to the previous study, they remain unknown to the reader of this paper. The previous work is published in Clinical Infectious Diseases which unlike Spectrum is not open access and likely not available to many (e.g., general practitioners) who might find the current study, including the limitations, of

interest. Authors should please take this into consideration.

2. One additional limitation that is not stated in this or the previous paper is the fact that urine culture and susceptibility tests are not performed on all patients with urinary tract infections that require therapy. This can have a significant impact on antibiogram data and should be understood by those who utilize antibiogram data for decision making. Please see <https://doi.org/10.1093/jac/dkl432>.

3. Authors reference CLSI M39 as a source for recommendations for compilation of antibiogram data. However, M39 suggests calculating percent susceptible and not percent resistant for the antibiogram. It would be helpful for authors to briefly explain why they elected to calculate and present percent resistant and include percent intermediate in the percent resistant statistics. According to definition, "the intermediate category implies clinical efficacy in body sites where the drugs are physiologically concentrated", which may apply to isolates in uncomplicated urinary tract infections. CLSI M100 27th ed (2017).

4. Line 103 - please add a reference for the IDSA guidelines

Staff Comments:

Preparing Revision Guidelines

Please return the manuscript within 60 days; if you cannot complete the modification within this time period, please contact me. If you do not wish to modify the manuscript and prefer to submit it to another journal, please notify me of your decision immediately so that the manuscript may be formally withdrawn from consideration by Microbiology Spectrum.

Dear Ahmed Babiker,

Thank you for the opportunity to submit a revised draft of our manuscript titled " Outpatient Antibiotic Resistance Patterns of Escherichia coli Urinary Isolates Differ by Specialty Type ". We appreciate the time and effort that you and the reviewers put in to provide feedback and insightful comments. We have incorporated changes to reflect the suggestions provided by the reviewers and highlights the changes within the manuscript.

Reviewer #1 (Comments for the Author):

Line 53 - I believe trimethoprim-sulfa would be considered an abbreviation for trimethoprim-sulfamethoxazole. Would write out the latter once.

Response: Trimethoprim-sulfamethoxazole is written out on line 53.

Lines 75-77 - would use this sentence to point out that associations were found, rather than just assessed

Response: The sentence has been changed to state that associations were found from the previous study.

Line 80 - Write out obstetrics and gynecology (Ob/Gyn) here rather than in line 96 -

Response: Obstetrics and gynecology is written out on line 81, with all other references using the abbreviated Ob/Gyn.

Line 109 - Missing a word - perhaps exist? "To assess whether differences exist..."

Response: The sentence has been corrected and "exist" has been added to the sentence in line 109.

Line 118 - Don't need to redefine Ob/Gyn abbreviation here

Response: Obstetrics and gynecology is written out on line 81, with all other references using the abbreviated Ob/Gyn.

Line 126 - I think "other" here refers to the "All other specialties" category. Would keep terminology consistent.

Response: "All other specialties" has replaced "other" in the sentence in line 125.

Line 133 - "Change difference" to "different"

Response: "Different" has replaced "difference" in the sentence in line 133.

Line 139 - Typically the lower number in the range would be written first

Response: The percentage has been corrected to include the lower range first.

Line 158 - Ob/gyn had higher odds of resistance for nitrofurantoin

Response: The sentence in line 159 has been corrected to give the correct direction of association.

Line 168 - If such urology-specific guidance exists, would cite it and point out how this data (and other data) suggests a need for revision

Response: Citations for the 2019 AUA guidelines has been added as well as an explanation.

Lines 170-173 - I think (and some of my personal bias from stewardship experience may be showing here) you are being too generous to the Urologists here. If they were aware of prior resistance patterns, this and other data would suggest that ciprofloxacin should not be the top antibiotic prescribed from Urology offices (as you cited from the NAMCS data). A plausible speculation here would be whether the frequent choice of ciprofloxacin by urologists could be driving the development of resistance in their patients. It also seems to suggest a clear opportunity for antimicrobial stewardship teams to intervene with education around best empiric antibiotic choice.

Response: The wording has changed to remove the sentence addressing urology specialists' awareness and highlighting the need for specialty-specific local antibiograms and antimicrobial stewardship interventions around educations and the best empiric antibiotic choice.

Line 173 - US guidelines would be more relevant to cite here than Korean guidelines

Response: AUA 2019 guidelines have been added as a reference.

Lines 182-183 - I believe the IDSA guidelines only discuss nonpregnant women

You are correct, the IDSA does not discuss nonpregnant women. The citation has been changed to reflect the true source of this recommendation from the American College of Obstetricians and Gynecologists.

Additional guidelines on treatment of UTIS for pregnant women have been added

Line 187 - With regard to limitations, the decision to only include the first isolate and rationale for this is discussed in the authors' prior cited publication, however may carry greater weight in this study. For example, a patient may have a first urine culture sent from family medicine prior to referral to Urology, and then no urine cultures would be attributed to Urology over the subsequent 5 years, if I understand the methodology correctly. This may also potentially explain the relatively low number of isolates for Urology practices (including only 40 isolates from male patients from 20 practices over 5 years). As such, revisions to the methodology such as including one isolate per year per patient, including the first isolate per patient per specialty over the time period, or even including all isolates could be a consideration, to determine if it may be more informative about the within-specialty resistance patterns. At minimum, would articulate this limitation within the text of this manuscript rather than only referring to the prior one.

Response: Additional explanation on study decisions for including the first patient isolate were added as well as addition information on the previous work of from the manuscript Frisbie et al 2021.

Line 198 - "CDCs" is missing an apostrophe

Response: An apostrophe was added to CDC.

Table 1

--88.8% of isolates being from female patients for Urology practices seems quite high, given typically the large majority of patients in Urology practices are male. This seems to warrant a 2nd look to ensure the data is accurate / Urology practices are accurately categorized.

Response: Urology clinics included facility types identified as "Nephrology" or "Urology" facilities. There were only a total of 357 patients who has isolates collected from a urology clinic, 40 being men and 317 being women.

Table 2

--Why does the number of isolates vary between antibiotics? Were different susceptibility panels used? Would offer an explanation of this somewhere, assuming the numbers are accurate.

Response: Susceptibility panels differed by patient, so there is not an equal distribution of antibiotics test for a specific antibiotic across the patient population. This explains the differing denominators in Table 2. A "Note" explaining these differences has been added to Table 2.

--There seems to be a discrepancy in the counting of isolates between this table and Table 1. For example, in Table 1, the total number of isolates for General family practice is listed as 17,252, but the number of isolates is higher than this for each antibiotic in Table 2. The count for "All other specialties" seems substantially lower in Table 2 compared to Table 1.

Response: Thank you for bringing this to our attention. After reviewing the code, there was an error that caused inaccurate tables counts. The model estimates are correct and the estimates in Table 2 remain the same. The counts of patients per antibiotic-facility in the "Count, n" column are now updated and correct.

Reviewer #2 (Comments for the Author):

The authors analyzed data obtained from antimicrobial susceptibility testing at a large clinical reference laboratory to determine resistance rates for E. coli isolates from outpatients seen at 7 different types of outpatient clinical practices. They were interested in seeing if segregating antibiograms by practice might be useful for antimicrobial stewardship. Agents commonly used to treat urinary tract infections are included in the evaluation. The study, which "builds off" a previous study, is well done and easy to comprehend.

1. The authors indicate in line 187 that the "this study builds off previous work from the main author and carries the same limitations." These limitations are quite interesting, however, unless one has access to the previous study, they remain unknown to the reader of this paper. The previous work is published in Clinical Infectious Diseases which unlike Spectrum is not open access and likely not available to many (e.g., general practitioners) who might find the current study, including the limitations, of interest. Authors should please take this into consideration.

Response: Additional explanation on study decisions for including the first patient isolate were added as well as addition information on the previous work of from the manuscript Frisbie et al 2021.

2. One additional limitation that is not stated in this or the previous paper is the fact that urine culture and susceptibility tests are not performed on all patients with urinary tract infections that require therapy. This can have a significant impact on antibiogram data and should be understood by those who utilize antibiogram data for decision making. Please see <https://doi.org/10.1093/jac/dkl432>.

Response: Thank you for pointing this out and for the helpful reference. Language bringing up this limitation has been added to lines 209-212.

3. Authors reference CLSI M39 as a source for recommendations for compilation of antibiogram data. However, M39 suggests calculating percent susceptible and not percent resistant for the antibiogram. It would be helpful for authors to briefly explain why they elected to calculate and present percent resistant and include percent intermediate in the percent resistant statistics. According to definition, "the intermediate category implies clinical efficacy in body sites where the drugs are physiologically concentrated", which may apply to isolates in uncomplicated urinary tract infections. CLSI M100 27th ed (2017).

Response: The outcome of resistance used in the model was defined by a AST results of “resistant” or “intermediate”. For the purposes of this study, we elected to look at the outcome of resistance, and in turn structured the manuscript to look at the data from this angle. The authors are not suggesting against the CLSI recommended presentation of antibiogram data in the format of percent susceptible.

4. Line 103 - please add a reference for the IDSA guidelines

Response: IDSA guidelines have been added to references.

April 4, 2022

Dr. Ann E Salm
Quest Diagnostics
Infectious Diseases/Immunology
500 Plaza Drive
Secaucus, NJ 07094

Re: Spectrum02373-21R1 (Outpatient Antibiotic Resistance Patterns of Escherichia coli Urinary Isolates Differ by Specialty Type)

Dear Dr. Ann E Salm:

I comment the authors on addressing the reviewer responses. I have a few minor comments

I agree that the distribution of gender among urology patients seems unusually skewed towards female. I would recommend the authors consider changing this category label to urology/nephrology given

Lines 190-191: "We included the first isolate recorded for each individual patient during the five-year period, as recommended by Clinical and Laboratory Standards Institute (CLSI) for analysis and presentation of cumulative antimicrobial susceptibility test data"

This maybe better placed in the methods section and may aid readability

Thank you for submitting your manuscript to Microbiology Spectrum. As you will see your paper is very close to acceptance. Please modify the manuscript along the lines I have recommended. As these revisions are quite minor, I expect that you should be able to turn in the revised paper in less than 30 days, if not sooner. If your manuscript was reviewed, you will find the reviewers' comments below.

When submitting the revised version of your paper, please provide (1) point-by-point responses to the issues I raised in your cover letter, and (2) a PDF file that indicates the changes from the original submission (by highlighting or underlining the changes) as file type "Marked Up Manuscript - For Review Only". Please use this link to submit your revised manuscript. Detailed instructions on submitting your revised paper are below.

Link Not Available

Sincerely,

Ahmed Babiker

Reviewer comments:

Preparing Revision Guidelines

To submit your modified manuscript, log onto the eJP submission site at <https://spectrum.msubmit.net/cgi-bin/main.plex>. Go to Author Tasks and click the appropriate manuscript title to begin the revision process. The information that you entered when you

first submitted the paper will be displayed. Please update the information as necessary. Here are a few examples of required updates that authors must address:

- point-by-point responses to the issues I raised in your cover letter
- Upload a compare copy of the manuscript (without figures) as a "Marked-Up Manuscript" file.
- Each figure must be uploaded as a separate file, and any multipanel figures must be assembled into one file.
- Manuscript: A .DOC version of the revised manuscript
- Figures: Editable, high-resolution, individual figure files are required at revision, TIFF or EPS files are preferred

Please return the manuscript within 60 days; if you cannot complete the modification within this time period, please contact me. If you do not wish to modify the manuscript and prefer to submit it to another journal, please notify me of your decision immediately so that the manuscript may be formally withdrawn from consideration by Microbiology Spectrum.

Editor Comments:

I commend the authors on addressing the reviewer responses. I have a two minor comments

Comment 1:

I agree with reviewer 1 that the distribution of gender among urology patients seems unusually skewed towards female. I would recommend the authors consider changing this category label to urology/nephrology given the methodology used to pull this data.

Comment 2:

Lines 190-191: "We included the first isolate recorded for each individual patient during the five-year period, as recommended by Clinical and Laboratory Standards Institute (CLSI) for analysis and presentation of cumulative antimicrobial susceptibility test data"

This maybe better placed in the methods section and may aid general readability of the paper.

Dear Ahmed Babiker,

Thank you for the opportunity to submit a revised draft of our manuscript titled " Outpatient Antibiotic Resistance Patterns of Escherichia coli Urinary Isolates Differ by Specialty Type ". We appreciate the time and effort that you and the reviewers put in to provide feedback and insightful comments. We have incorporated changes to reflect the suggestions provided by the reviewers and highlights the changes within the manuscript.

Reviewer #1 (Comments for the Author):

Line 53 - I believe trimethoprim-sulfa would be considered an abbreviation for trimethoprim-sulfamethoxazole. Would write out the latter once.

Response: Trimethoprim-sulfamethoxazole is written out on line 53.

Lines 75-77 - would use this sentence to point out that associations were found, rather than just assessed

Response: The sentence has been changed to state that associations were found from the previous study.

Line 80 - Write out obstetrics and gynecology (Ob/Gyn) here rather than in line 96 -

Response: Obstetrics and gynecology is written out on line 81, with all other references using the abbreviated Ob/Gyn.

Line 109 - Missing a word - perhaps exist? "To assess whether differences exist..."

Response: The sentence has been corrected and "exist" has been added to the sentence in line 109.

Line 118 - Don't need to redefine Ob/Gyn abbreviation here

Response: Obstetrics and gynecology is written out on line 81, with all other references using the abbreviated Ob/Gyn.

Line 126 - I think "other" here refers to the "All other specialties" category. Would keep terminology consistent.

Response: "All other specialties" has replaced "other" in the sentence in line 125.

Line 133 - "Change difference" to "different"

Response: "Different" has replaced "difference" in the sentence in line 133.

Line 139 - Typically the lower number in the range would be written first

Response: The percentage has been corrected to include the lower range first.

Line 158 - Ob/gyn had higher odds of resistance for nitrofurantoin

Response: The sentence in line 159 has been corrected to give the correct direction of association.

Line 168 - If such urology-specific guidance exists, would cite it and point out how this data (and other data) suggests a need for revision

Response: Citations for the 2019 AUA guidelines has been added as well as an explanation.

Lines 170-173 - I think (and some of my personal bias from stewardship experience may be showing here) you are being too generous to the Urologists here. If they were aware of prior resistance patterns, this and other data would suggest that ciprofloxacin should not be the top antibiotic prescribed from Urology offices (as you cited from the NAMCS data). A plausible speculation here would be whether the frequent choice of ciprofloxacin by urologists could be driving the development of resistance in their patients. It also seems to suggest a clear opportunity for antimicrobial stewardship teams to intervene with education around best empiric antibiotic choice.

Response: The wording has changed to remove the sentence addressing urology specialists' awareness and highlighting the need for specialty-specific local antibiograms and antimicrobial stewardship interventions around educations and the best empiric antibiotic choice.

Line 173 - US guidelines would be more relevant to cite here than Korean guidelines

Response: AUA 2019 guidelines have been added as a reference.

Lines 182-183 - I believe the IDSA guidelines only discuss nonpregnant women

You are correct, the IDSA does not discuss nonpregnant women. The citation has been changed to reflect the true source of this recommendation from the American College of Obstetricians and Gynecologists.

Additional guidelines on treatment of UTIS for pregnant women have been added

Line 187 - With regard to limitations, the decision to only include the first isolate and rationale for this is discussed in the authors' prior cited publication, however may carry greater weight in this study. For example, a patient may have a first urine culture sent from family medicine prior to referral to Urology, and then no urine cultures would be attributed to Urology over the subsequent 5 years, if I understand the methodology correctly. This may also potentially explain the relatively low number of isolates for Urology practices (including only 40 isolates from male patients from 20 practices over 5 years). As such, revisions to the methodology such as including one isolate per year per patient, including the first isolate per patient per specialty over the time period, or even including all isolates could be a consideration, to determine if it may be more informative about the within-specialty resistance patterns. At minimum, would articulate this limitation within the text of this manuscript rather than only referring to the prior one.

Response: Additional explanation on study decisions for including the first patient isolate were added as well as addition information on the previous work of from the manuscript Frisbie et al 2021.

Line 198 - "CDCs" is missing an apostrophe

Response: An apostrophe was added to CDC.

Table 1

--88.8% of isolates being from female patients for Urology practices seems quite high, given typically the large majority of patients in Urology practices are male. This seems to warrant a 2nd look to ensure the data is accurate / Urology practices are accurately categorized.

Response: Urology clinics included facility types identified as "Nephrology" or "Urology" facilities. There were only a total of 357 patients who has isolates collected from a urology clinic, 40 being men and 317 being women.

Table 2

--Why does the number of isolates vary between antibiotics? Were different susceptibility panels used? Would offer an explanation of this somewhere, assuming the numbers are accurate.

Response: Susceptibility panels differed by patient, so there is not an equal distribution of antibiotics test for a specific antibiotic across the patient population. This explains the differing denominators in Table 2. A "Note" explaining these differences has been added to Table 2.

--There seems to be a discrepancy in the counting of isolates between this table and Table 1. For example, in Table 1, the total number of isolates for General family practice is listed as 17,252, but the number of isolates is higher than this for each antibiotic in Table 2. The count for "All other specialties" seems substantially lower in Table 2 compared to Table 1.

Response: Thank you for bringing this to our attention. After reviewing the code, there was an error that caused inaccurate tables counts. The model estimates are correct and the estimates in Table 2 remain the same. The counts of patients per antibiotic-facility in the "Count, n" column are now updated and correct.

Reviewer #2 (Comments for the Author):

The authors analyzed data obtained from antimicrobial susceptibility testing at a large clinical reference laboratory to determine resistance rates for E. coli isolates from outpatients seen at 7 different types of outpatient clinical practices. They were interested in seeing if segregating antibiograms by practice might be useful for antimicrobial stewardship. Agents commonly used to treat urinary tract infections are included in the evaluation. The study, which "builds off" a previous study, is well done and easy to comprehend.

1. The authors indicate in line 187 that the "this study builds off previous work from the main author and carries the same limitations." These limitations are quite interesting, however, unless one has access to the previous study, they remain unknown to the reader of this paper. The previous work is published in Clinical Infectious Diseases which unlike Spectrum is not open access and likely not available to many (e.g., general practitioners) who might find the current study, including the limitations, of interest. Authors should please take this into consideration.

Response: Additional explanation on study decisions for including the first patient isolate were added as well as addition information on the previous work of from the manuscript Frisbie et al 2021.

2. One additional limitation that is not stated in this or the previous paper is the fact that urine culture and susceptibility tests are not performed on all patients with urinary tract infections that require therapy. This can have a significant impact on antibiogram data and should be understood by those who utilize antibiogram data for decision making. Please see <https://doi.org/10.1093/jac/dkl432>.

Response: Thank you for pointing this out and for the helpful reference. Language bringing up this limitation has been added to lines 209-212.

3. Authors reference CLSI M39 as a source for recommendations for compilation of antibiogram data. However, M39 suggests calculating percent susceptible and not percent resistant for the antibiogram. It would be helpful for authors to briefly explain why they elected to calculate and present percent resistant and include percent intermediate in the percent resistant statistics. According to definition, "the intermediate category implies clinical efficacy in body sites where the drugs are physiologically concentrated", which may apply to isolates in uncomplicated urinary tract infections. CLSI M100 27th ed (2017).

Response: The outcome of resistance used in the model was defined by a AST results of “resistant” or “intermediate”. For the purposes of this study, we elected to look at the outcome of resistance, and in turn structured the manuscript to look at the data from this angle. The authors are not suggesting against the CLSI recommended presentation of antibiogram data in the format of percent susceptible.

4. Line 103 - please add a reference for the IDSA guidelines

Response: IDSA guidelines have been added to references.

April 4th: Additional feedback from reviewers

Comment 1:

I agree with reviewer 1 that the distribution of gender among urology patients seems unusually skewed towards female. I would recommend the authors consider changing this category label to urology/nephrology given the methodology used to pull this data.

Response: agree and have changed all references of “urology” as a practice to “urology/nephrology”. Additionally, an abbreviation of “uro-neph” has been explained/incorporated throughout the paper to ensure consistency on noting this practice type.

Comment 2:

Lines 190-191: "We included the first isolate recorded for each individual patient during the five-year period, as recommended by Clinical and Laboratory Standards Institute (CLSI) for analysis and presentation of cumulative antimicrobial susceptibility test data"

This maybe better placed in the methods section and may aid general readability of the paper.

Response: agree and have made this change.

April 11, 2022

Dr. Ann E Salm
Quest Diagnostics
Infectious Diseases/Immunology
500 Plaza Drive
Secaucus, NJ 07094

Re: Spectrum02373-21R2 (Outpatient Antibiotic Resistance Patterns of Escherichia coli Urinary Isolates Differ by Specialty Type)

Dear Dr. Ann E Salm:

Thank you for addressing the latest round of comments. Your manuscript has been accepted, and I am forwarding it to the ASM Journals Department for publication. You will be notified when your proofs are ready to be viewed.

Sincerely,

Ahmed Babiker
Editor, Microbiology Spectrum
